# Coppice Management for Young Sycamore Maple (*Acer pseudoplatanus* L.)

**Bogdan M. Strimbu** [1,2] and **Valeriu-Norocel Nicolescu** [3,*]

1   College of Forestry, Oregon State University, 3100 Jefferson Way, Corvallis, OR 97331, USA
2   National Institute of Research and Development for Biological Sciences, 296 Independenței Bd, Sector 6, 060031 Bucharest, Romania
3   Faculty of Silviculture and Forest Engineering, Transylvania University of Brasov, Sirul Beethoven 1, 500123 Brasov, Romania
*   Correspondence: nvnicolescu@unitbv.ro

**Abstract:** Sycamore is a valuable tree not only economically but also ecological and culturally. Even though it has a vigorous regeneration system from its stump, its coppice management has triggered limited formal investigations. Therefore, the present study focused on finding the most suitable coppice strategy for achieving ground coverage and biomass, as well as developing growth and yield models for sycamore maples. Using a series of eight measurements spanning twenty-one years, starting from age six, we found that single-shoot coppices provided superior yields for height than seed-managed trees up to age twelve and up to age twenty for DBH. The coppice trees outperformed the seed trees up to age 10. The yield of DBH and the height for single-shoots and seed-managed trees were described by parsimonious formulations, namely the Schumacher model for DBH and the square root for height. The relationship of DBH–height exhibited a clear linear form, pointing toward the main limitation of the study, namely the confinement to ages less than 20 years. Nevertheless, all the models exhibited a bias $<10^{-7}$ and an $R^2$ around 80%, except for the height and DBH change throughout time, which was around 67%.

**Keywords:** high forest; number of shoots; diameter–height modeling; mixed model; Romania

## 1. Introduction

Naturally regenerated forests (i.e., 3.75 billion hectares) account for ninety-three percent of the world's forest area (i.e., 4.06 billion hectares; thirty-one percent of the Earth's land surface), while only seven percent of the forest area worldwide (i.e., 271 million hectares) is composed of plantations [1,2]. Forests regenerated by seed, either naturally or artificially (so-called high forests), are predominant, and only a small share of the world's forests are propagated vegetatively (by stump shoots, stool shoots and root suckers), which are labeled by Nyland [3] as low-forest. The main silvicultural system applied to low-forests is coppice [3].

In Europe, where forests cover about 227 million ha (35 percent of total land area) [4], coppices occupy an area of about 23 million hectares, comprising a little over a tenth of the continent's forest area [5]. Simple coppice, which is a forest management system in which trees are systematically and repetitively cut and regeneration is vegetative by means of sprouting or suckering, is applied especially in broadleaved tree species that can withstand repeated cutting, such as oaks (*Quercus* spp.), sweet chestnut (*Castanea sativa* L.), hornbeam (*Carpinus betulus* L.), linden (*Tilia* spp.), ash (*Fraxinus* spp.), alders (*Alnus* spp.), black locust (*Robinia pseudoacacia* L.), poplars (*Populus* spp.), and eucalypts (*Eucalyptus* spp.) [5].

The coppice management system is not commonly used for sycamore maple (*Acer pseudoplatanus* L.), the most important maple species in Europe [6]. Sycamore maple is one of the most valuable broadleaved tree species on the continent, where it covers 1.2 percent of the forest land area and 1.7 percent of the total wood volume harvested annually [7].

Sycamore maple is rarely found growing in pure stands, is usually a component of mixed woodlands and is found in small groups or integrated with other tree species [8]. The wood of sycamore maple is hard and strong, and it is widely used for furniture, fine joinery or flooring. Figured sycamore maple (i.e., "wavy-grained", "fiddle-back") is exceptionally valuable for cutting into decorative veneers as well as for musical instruments of the violin family [8–18]. Other uses of the wood are sawn and pulpwood or even for firewood [11,15,17]. The juvenile growth of high forest sycamore maples is faster than that of most European broadleaved tree species, with it increasing in height by more than 1 m annually in the most productive soils [6,15,18,19].

Usually, sycamore maple is managed via high forest management [5], targeting the production of large-diameter trees. However, as with all maple species, sycamore maple can be coppiced as it regenerates vigorously [10,12,17], growing by as much as 1.5–2.5 m in the first year. Its rapid height growth combined with its large number of shoots leads to canopy closure occurring in the second or third year after coppicing [11]. The high yield of the young shoots promotes the coppice, and more than 2500 ha are present in Britain, and they experience a rotation of 10–20 years [11]). A property of sycamore maples that recommends a low-forest management style is their ability to coppice well up to 80–100 years old [8].

As it grows mostly in mixed stands, limited work has been carried out on the growth and yield modeling of sycamore maple. However, as a high-forest tree, sycamore's current annual increment (CAI) of height often culminates between age 12 and 14, while its mean annual increment of height (MAI) reaches its peak mostly between the 20th and 25th years of age [19,20], which recommends the development of accurate and precise models. Beyond age 50 years, the height growth decreases considerably [14,19]. In terms of diameter, the CAI peaks at age 20 to 40 years, depending on the site productivity, while the MAI culminates between the 30th and 50th years of age [21]. In sycamore maple, as its wood is diffusely porous, its ring width has little influence on its wood density (630–640 kg m$^{-3}$ at 12–17 per cent moisture content [19]). This means that its growth rate will not affect its strength properties [19].

Unfortunately, among the small number of papers dedicated to these coppices, many have a questionable scientific foundation, as no formal evidence has been provided to support the findings [20]. In addition, the usage of a single mathematical model to represent all the species irrespective of the regeneration system is likely to be inappropriate [22,23]. Furthermore, some models that describe the change in the species over time are prone to collinearity [24], as attributes that exhibit correlations larger than 80% are used as predictors [22,25]. Therefore, to address the lack of reliable information associated to coppice systems applied to sycamore trees, we aimed in this study to provide support for the selection of a particular management strategy. Additionally, we aimed to describe the dynamics of sycamore maples, and we developed growth and yield models for the change in the height and diameter at breast height (DBH) over time. To enhance the study, we compared coppiced sycamore maples with ones regenerated from seed growing on the same site.

## 2. Materials and Methods

### 2.1. Study Area and Experiment Establishment

To support our findings, we executed an experiment located in the Southern Carpathians (45°54′35″ N, 25°54′16″ E, mean elevation 780 m a.s.l.) in Romania (Figure 1). According to the Köppen–Geiger classification system, the climate in the studied region was of a D.f.k type (i.e., a continental climate with a mean annual temperature of 7.6 °C and a mean precipitation of 584.1 mm yr$^{-1}$). The site was relatively horizontal with a slope of less than 5°, and it had a brown argillic soil [26,27] with moderate fertility for natural vegetation consisting of a sessile oak-dominated mixture of broadleaves.

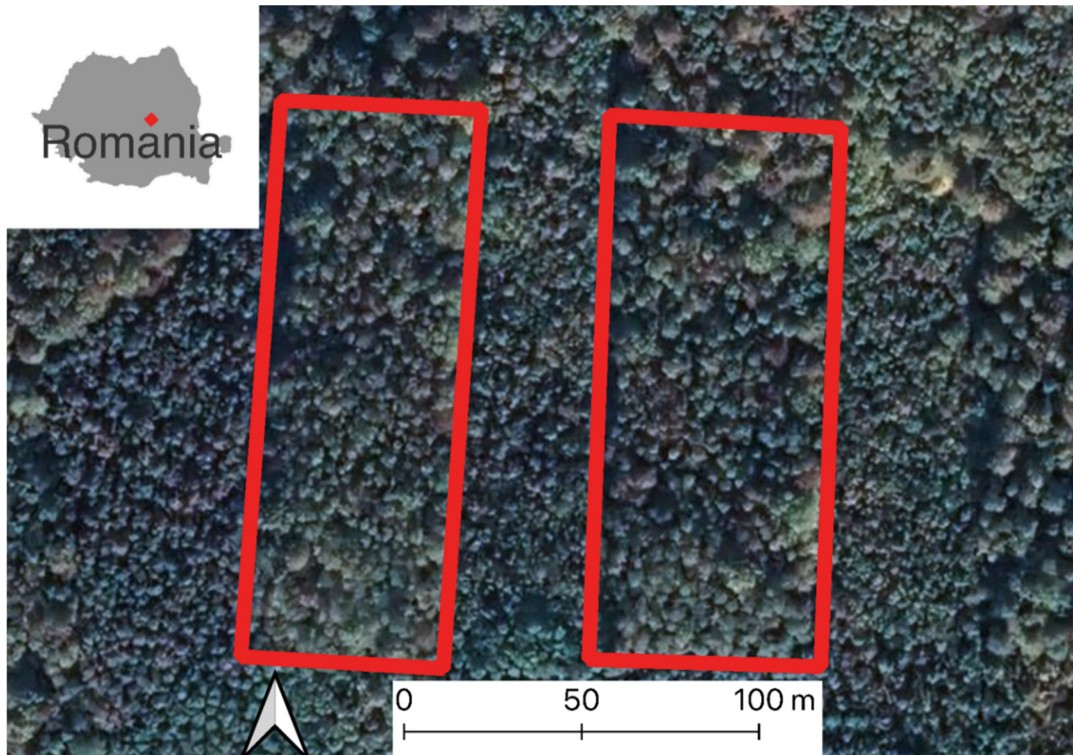

**Figure 1.** Study area in 2022. The red rectangle shows the sections on which the measurements were carried out, whereas the red dot in the inset points to the location of the study area within the region, east of Covasna County.

The trees were planted in April 2003, with one-year old seedlings being planted at a density of 5000 seedlings ha$^{-1}$ in a rectangular pattern of 1.8 m × 1.1 m. The species composition at planting was 52% northern red oak (*Quercus rubra* L.), 24% European beech (*Fagus sylvatica* L.), 15% sycamore maple and 9% European larch (*Larix decidua* Mill.). However, to fill the gaps created by dead seedlings, subsequent plantation with sessile oak (*Quercus petraea* (Matt.) Liebl.), small-leaved linden (*Tilia cordata* Mill.), wild cherry (*Prunus avium* L.), common walnut (*Juglans regia* L.), Norway spruce (*Picea abies* (L.) Karst), wild pear (*Pyrus pyraster* (L.) Burgsd.), silver birch (*Betula pendula* Roth) and horse chestnut (*Aesculus hippocastanum* L.) was carried out. The stand was managed from 2003 to 2008 as an agroforestry system, with strawberries being cultivated in the rows between the trees (Figure 2). The inclusion of a crop with an annual rotation not only ensured a steady income but also eliminated competition from other species and ensured the consistency of the stand density overtime.

*2.2. Selection of Number of Shoots*

The fast height growth of the sycamore maples shaded the strawberries; therefore, the rows of pure sycamore trees were cut at 5–10 cm above the collar in 2005. In the spring of 2011 (i.e., April and May), 40 stumps were cut selectively. On 15 stumps, only 1 shoot was left, on 15 stumps, 2 shoots were left, and no cut was made on 10 stumps (Figure 3). The stumps with no cuts, which served as controls, had between 4 and 7 shoots.

For all the shoots on the 40 stumps, we measured the diameter at breast height (DBH) at 8 ages, which was expressed at months 73, 96, 103, 114, 122, 133, 153 and 207. For the first five measurements, we also acquired the total height. We did not measure the total height starting at age 133 months because the crowns were very close and dense, which prevented the accurate identification of the top of each shoot. There were two main reasons for the irregular acquisition of data, which was conducted almost annually for the first six

times and with more than two years between measurements for the last two measurements. First, the landowner's focus was not on the trees but on the annual crop; therefore, all the measurements were executed to accommodate the main land management objectives. While the trees were relatively short, they were measured every year when the weather permitted. The last two measurements, however, were carried out opportunistically because there were no annual crops or external funding to support or justify the field effort.

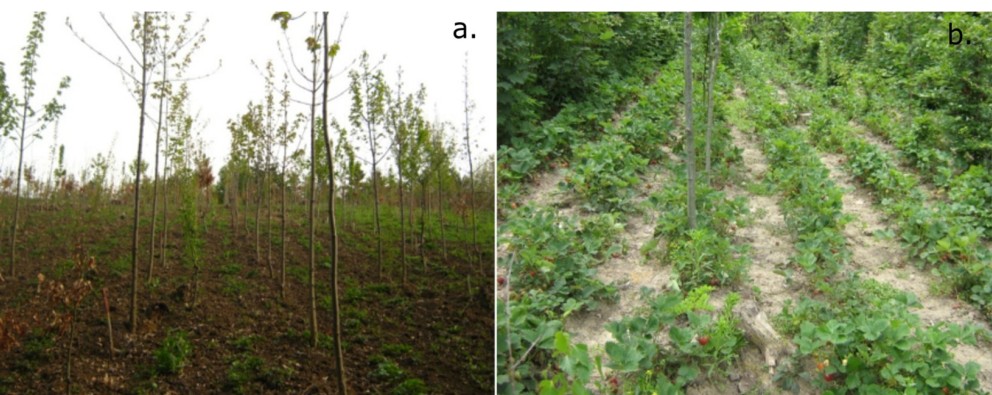

**Figure 2.** Agroforestry system used for the first 5 years of the stand. (**a**) Spring,(**b**) summer.

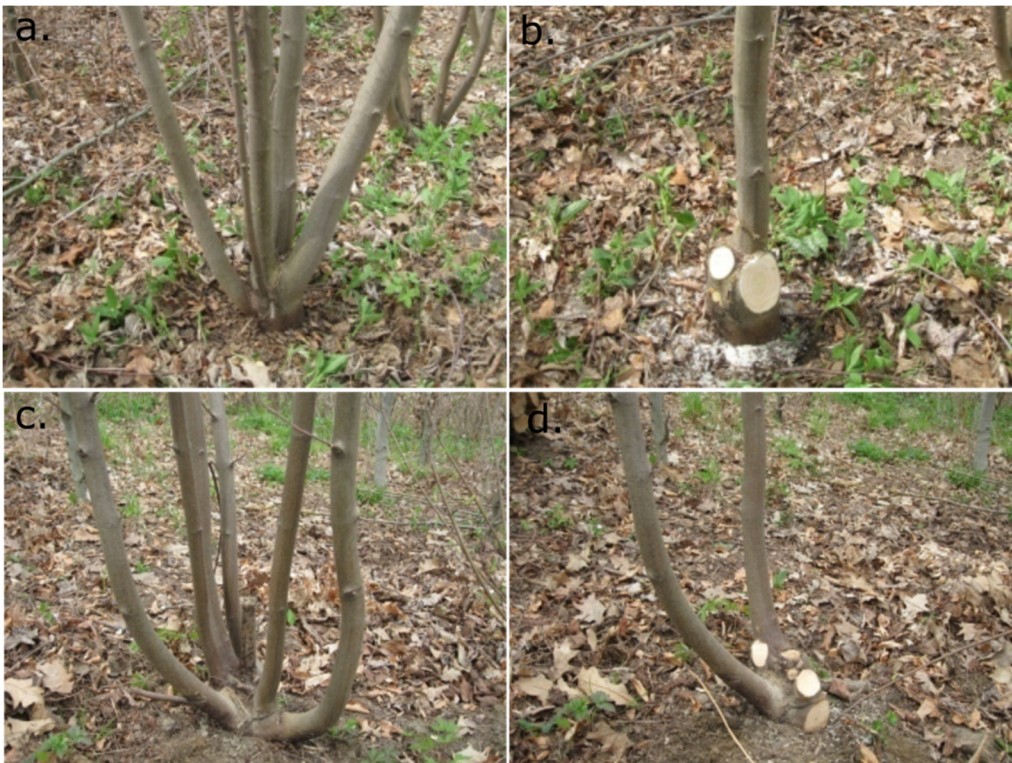

**Figure 3.** Treatment of stumps with one shoot ((**a**) before cut and (**b**) after cut) and two shoots ((**c**) before cut and (**d**) after cut).

In addition to the trees that were treated in the coppice system, we selected a set of 100 sycamore maples that were not cut; therefore, the sample was treated as a high forest. These trees, henceforth called seed-managed trees, served as a comparison for the shoots of the 40 stumps. Depending on the context, stems could either refer to shoots or seed-trees. Because of the slower initial growth, the seed-managed trees were measured 7 times from 122 months to 243 months at various intervals (i.e., 122, 139, 150, 158, 189 and 243 months).

The first five measurements included DBH and total height, whereas the last two included only DBH. The same argument for the unequal measurements presented for the shoots is valid for the seed-managed trees.

### 2.3. Growth and Yield Modeling

To identify the best coppice management strategy for the sycamore maple, we used a nested repeat measurements linear mixed model, as suggested by Diggle et al. [28] and Crowder and Hand [29]. The fixed effects were the number of shoots on the stump, and the random effects were the stump and the shoot, which was nested within the stump (Equation (1)).

$$\sqrt{y_{ijkl}} = SPS_i + Stump_j + Shoot_{k(j)} + e_l \tag{1}$$

where:

$SPS_i$ is the number of shoots of stump *I*;
$Stump_j$ is the stump *j*;
$Shoot_{k(j)}$ is the shoot *k* nested in the stump *j*;
$e_l$ is the residual of shoot *k* from stump *j* that has *i* shoots.

We tested the assumptions required for the nested repeat measurements linear mixed model using the Kolmogorov–Smirnov and Anderson–Darling tests for normality and the Durbin–Watson autocorrelation test, and heteroskedasticity was measured using an exponential link function. We tested three possible covariance matrix structures of the repeated measurements models, as suggested by Diggle et al. [28], namely compound symmetry, variance components and first-order autoregression. We assessed the difference among the number of shoots per stump using the Scheffe multiple comparison tests [30] developed on a least-squares means.

To enhance the analysis, we developed a yield model for the DBH and height for the shoots that provided the fastest development, namely the stumps with one shoot. We estimated the parameters of the model using the Gauss–Newton method and a convergence value of $10^{-5}$. We selected the final model using four criteria in the following order: significance (i.e., <0.001 or above), Akaike information criterion (i.e., the smaller the better), bias (i.e., smallest was preferred) and the coefficient of correlation between the predicted and the measured values (i.e., the larger the better). The final model was tested for the normality of the residuals, autocorrelation and homoskedasticity. In the eventuality that the model failed the independence test or the distribution was severely skewed or bimodal, then a different model was developed. All the computations were executed in SAS ver. 9.3 [31].

## 3. Results

The summary statistics describing the shoots and seed-managed trees revealed a wide range of DBHs and heights (Table 1). As expected, the DBH decreased with the number of shoots per stump (i.e., at least 1.5 mm), with a difference that increased over time (i.e., 4.1 cm at age 207 months). Similarly to other studies [5,6], the DBH disparity among the number of shoots was larger for the extreme values. The median DBH was constantly less than the mean DBH, providing a right-skewed distribution overtime. A similar picture was provided by the total height, but only for the shoots growing alone on a stump, as those with multiple shoots exhibited a reduced growth. Nevertheless, the height distribution seemed to be left-skewed, as the median was larger than the mean regardless of the age. In addition to the difference among the number of shoots on the stump, there was also a difference between the larger stems and smaller stems, as the range between the first and third quartile increased with the number of shoots on the stump (Figure 4).

**Table 1.** Summary statistics describing the change in DBH and height overtime.

| No. of Shoots | Age [Months] | DBH [cm] | | | | Total Height [m] | | | |
|---|---|---|---|---|---|---|---|---|---|
| | | Mean | Minimum | Maximum | Median | Mean | Minimum | Maximum | Median |
| 1 | 73 | 4.11 | 2.60 | 6.80 | 3.90 | 5.34 | 3.99 | 6.67 | 5.42 |
| | 96 | 5.42 | 3.40 | 7.90 | 5.20 | 6.27 | 5.00 | 7.05 | 6.30 |
| | 103 | 6.25 | 4.00 | 9.10 | 6.10 | 6.96 | 5.60 | 7.90 | 7.00 |
| | 114 | 6.79 | 4.10 | 10.20 | 6.60 | 7.31 | 5.90 | 8.35 | 7.20 |
| | 122 | 6.79 | 4.10 | 10.20 | 6.60 | 7.41 | 6.05 | 8.45 | 7.28 |
| | 133 | 7.24 | 4.10 | 11.00 | 7.00 | | | | |
| | 153 | 8.03 | 4.20 | 12.60 | 7.70 | | | | |
| | 207 | 9.45 | 4.30 | 15.70 | 9.40 | | | | |
| 2 | 73 | 3.94 | 2.40 | 6.00 | 3.90 | 5.19 | 4.05 | 6.68 | 5.11 |
| | 96 | 4.92 | 3.40 | 7.00 | 4.60 | 6.13 | 4.70 | 7.05 | 6.40 |
| | 103 | 5.37 | 3.60 | 7.80 | 5.05 | 6.65 | 5.21 | 7.62 | 6.82 |
| | 114 | 5.66 | 3.70 | 8.40 | 5.40 | 6.90 | 5.55 | 7.93 | 7.06 |
| | 122 | 5.66 | 3.70 | 8.40 | 5.40 | 6.98 | 5.60 | 8.03 | 7.15 |
| | 133 | 5.87 | 3.70 | 8.80 | 5.45 | | | | |
| | 153 | 6.40 | 3.90 | 10.00 | 5.85 | | | | |
| | 207 | 7.21 | 3.90 | 11.40 | 6.55 | | | | |
| >2 | 73 | 3.38 | 1.50 | 5.70 | 3.30 | 5.29 | 3.51 | 6.83 | 5.27 |
| | 96 | 3.89 | 2.00 | 6.00 | 4.00 | 6.07 | 4.20 | 7.30 | 6.15 |
| | 103 | 4.16 | 2.00 | 6.40 | 4.30 | 6.56 | 4.35 | 8.05 | 6.67 |
| | 114 | 4.23 | 2.00 | 6.60 | 4.30 | 6.84 | 4.35 | 8.45 | 7.14 |
| | 122 | 4.23 | 2.00 | 6.60 | 4.30 | 6.91 | 4.40 | 8.55 | 7.21 |
| | 133 | 4.43 | 2.80 | 6.90 | 4.50 | | | | |
| | 153 | 4.79 | 2.80 | 7.80 | 4.70 | | | | |
| | 207 | 5.23 | 3.30 | 9.30 | 5.05 | | | | |
| Seed | 109 | 5.67 | 3.90 | 8.10 | 5.75 | 5.34 | 4.07 | 6.45 | 5.33 |
| | 122 | 6.60 | 4.40 | 9.00 | 6.65 | 5.80 | 4.62 | 7.17 | 5.85 |
| | 139 | 7.92 | 5.30 | 10.80 | 7.90 | 6.37 | 5.20 | 7.61 | 6.38 |
| | 150 | 8.44 | 5.50 | 11.60 | 8.45 | 6.63 | 5.40 | 7.81 | 6.69 |
| | 158 | 8.45 | 5.50 | 11.60 | 8.45 | 6.76 | 5.44 | 7.95 | 6.78 |
| | 189 | 9.84 | 6.00 | 13.40 | 10.10 | | | | |
| | 243 | 11.45 | 6.00 | 15.10 | 11.60 | | | | |

The seed-managed trees revealed that the DBH caught up with the coppice trees by age 9 for samples with multiple shoots and by age 11 for samples with single shoots, after which the growth accelerated compared with the shoots. A different picture was portrayed by the height, which did not catch up with the shoots even after 156 months, regardless of the number of stems per stump.

For ages less than 144 months, the DBH and height seemed to be linearly related (Figure 5). As pointed by other studies [5,6], the larger the number of shoots on a stump, the slower the dimensional growth. It should be noted that there was a lack of an asymptote for both the DBH and height, which suggested that the trees were still in the active growth period of their life.

The nested repeat measurements linear mixed model used to analyze the data enhanced the findings provided by the summary statistics by showing that the DBH differed significantly among the number of shoots left on the stump (Table 2). When the shoots were compared only among themselves, the Scheffe multiple comparison test suggested that the number of stems differed significantly, regardless of the number of shoots (i.e., $p$-value < 0.0068). When the comparison included the seed-managed trees, the stumps with only one shoot were significantly different to the rest, including the seed-managed trees (i.e., $p$-value = 0.0072). The analysis of the height was executed only for the shoots, as there were too few measurements to warrant a meaningful inference that included the seed-managed trees (two overlapping measurements). The analysis revealed that the height did not depend on the number of shoots left on the stump, as the $p$-value > 0.54 (Table 2).

The first-order autoregression structure of the covariance matrix provided the smallest AIC, which was used in all the analyses (i.e., 2811.2 for DBH and 1205.9 for height). The Durbin–Watson test did not suggest the presence of correlated residuals ($p$-value $> 0.1$), but the residuals were not normally distributed or heteroskedastic (i.e., $p$-value $< 0.001$). However, the distribution of residuals was unimodal and had a limited skew, which supported the findings, as the lack of the two assumptions reduced the power of the test rather than invalidating it.

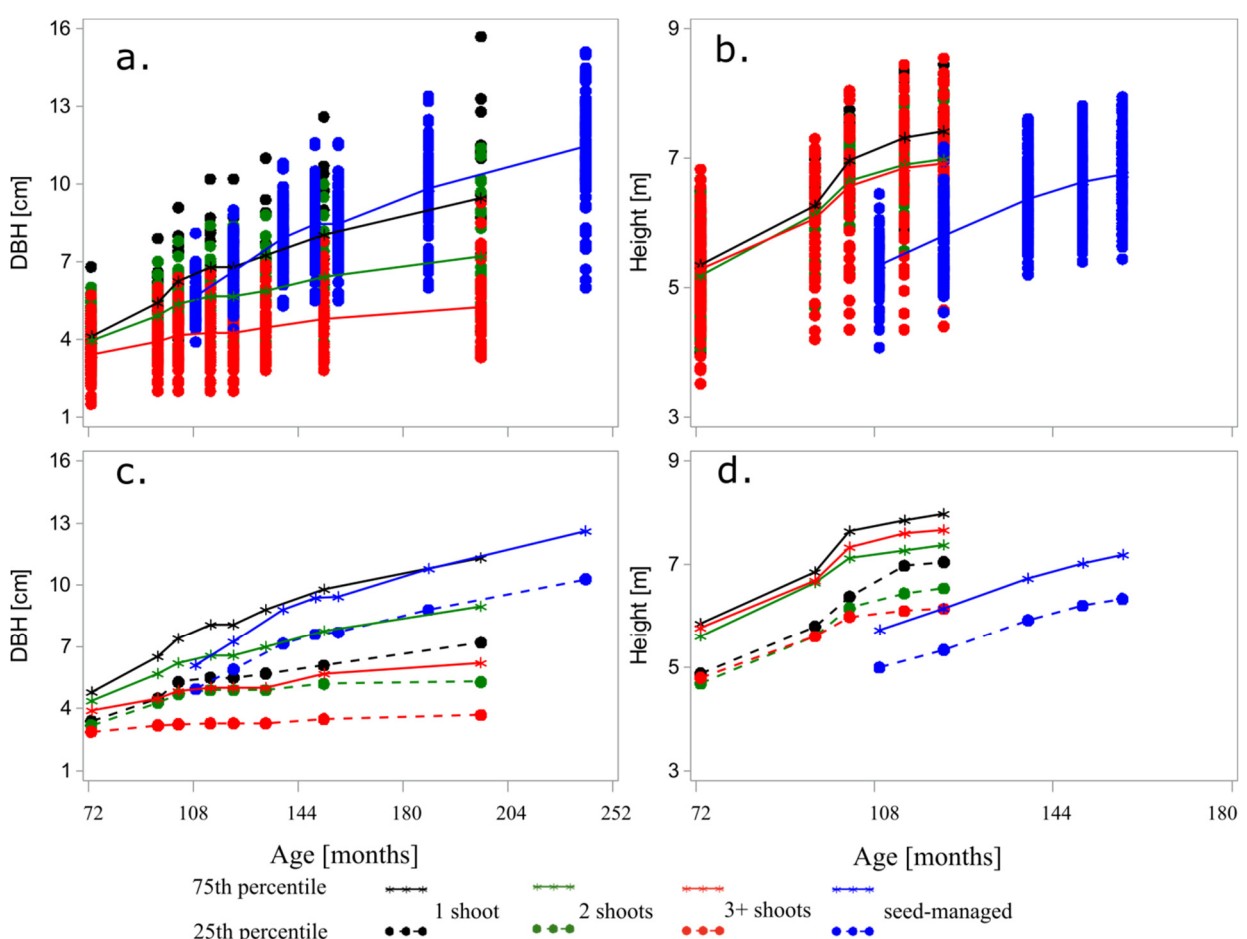

**Figure 4.** Change overtime of (**a**) DBH, (**b**) total height, (**c**) DBH in first and third quartile and (**d**) height in first and third quartile. Solid line with no symbols indicates the mean value.

**Table 2.** $p$-values of the Scheffe multiple comparison tests for DBH (above main diagonal) and height (below main diagonal) provided by the nested repeat measurements linear mixed model of the DBH and total height.

| No. of Stems/Stump | 1 | 2 | >2 | Seed |
|---|---|---|---|---|
| 1 | - | 0.0068 | <0.001 | 0.0072 |
| 2 | 0.69 | - | <0.001 | <0.001 |
| >2 | 0.54 | 0.97 | - | <0.001 |

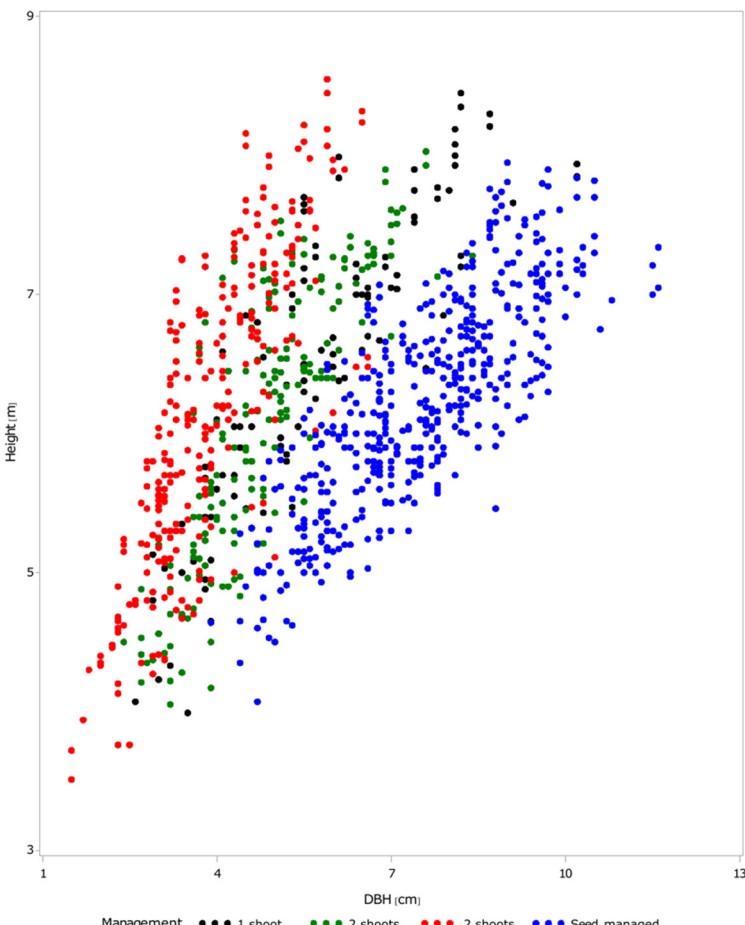

**Figure 5.** DBH–height relationship for stems younger than 144 months for the four management strategies: coppice with one shoot (red), coppice with two shoots (green), coppice with three or more shoots (black) and seed-managed trees (blue).

Given the results supplied by the nested repeated measurements, we only developed models for the single stems (i.e., stumps with one shoot and seed-managed trees), as they provided comparable dimensions, irrespective of the management system (i.e., coppice or high forest). Among the non-linear models tested for DBH, the Schumacher model [32] was the most parsimonious [33], and it exhibited suitable assessment metrics (i.e., AIC, bias, coefficient of correlation of the predicted and measured data, autocorrelation, normality and heteroskedasticity) (Equation (2).

$$D\hat{B}H = b_0 + b_1 e^{b_2/age} \tag{2}$$

where:

$b_0$ is −0.7125 for single-shoot trees and −9.6706 for seed-managed trees;
$b_1$ is 15.1450 for single-shoot trees and 27.2421 for seed-managed trees;
$b_2$ is −6.9608 for single-shoot trees and −5.1995 for seed-managed trees.

For the height, we also tried using a large series of models, with the square root model (Equation (3)) outperforming the rest of the considered equations (e.g., exponential, Schumacher, Chapman–Richards, Korf, sine, cosine, tangent, arcsine, arctangent, quadratic and cubic). Similar results to the DBH were obtained for the height, with some assumptions being met and others being violated. Nevertheless, the failed assumptions did not affect the findings, as the distribution of the residuals was unimodal and relatively symmetrical,

pointing to a reduced ability to detect changes due to chance rather than by a computational or conceptual issue.

$$\hat{height} = b_0 + b_1\sqrt{age} \tag{3}$$

where:

$b_0$ is $-2.1416$ for single-shoot trees and $-1.6315$ for seed-managed trees;
$b_1$ is 3.0358 for single-shoot trees and 2.3298 for seed-managed trees.

We developed two inverse relationships for the DBH and height, because various applications require either a prediction of the height from the DBH or vice versa (Table 3). Nevertheless, the functions are not mathematically the inverse of each other, because both relationships have an asymptote with each other (Table 3). The need for two relationships is also advocated by the dependency of the predicted variable on the instrument, as devices in direct contact with trees usually measure the DBH [34], whereas remote sensing devices such as LiDAR commonly estimate the total height [35]. Therefore, we developed two equations, one predicting the height from the DBH, which is suitable for ground measurements, and one predicting the DBH from the height, which is suitable for aerial estimates (Table 3). The linear relationships inferred from are as follows:

$$\hat{DBH} = b_0 + b_1 \times height \tag{4}$$

$$\hat{Height} = b_0 + b_1 \times DBH \tag{5}$$

where for Equation (4):

$b_0$ is $-3.25453460$ for single-shoot trees and $-2.55698392$ for seed-managed trees;
$b_1$ is 1.371592838 for single-shoot trees and 1.619382533 for seed-managed trees.

For Equation (5):

$b_0$ is 3.459652409 for single-shoot trees and 3.263246479 for seed-managed trees;
$b_1$ is 0.536159811 for single-shoot trees and 0.394919299 for seed-managed trees.

**Table 3.** Assessment statistics for the DBH and height models. The units for bias and AIC are defined by the model, with cm being the unit for DBH and meters being the unit for height. Signif. represents the significance of the model as computed using the F-test and Independ. stands for the Durbin–Watson autocorrelation tests.

| Model | Signif. | Bias | Corr. Coeff. | AIC | Independ. | Normal | Hetero. |
|---|---|---|---|---|---|---|---|
| Single-shoot/seed-managed | *p*-value | [model] | % | [model] | *p*-value | *p*-value | *p*-value |
| Equation(2): DBH = f(age) | <0.001 | $-2 \times 10^{-7}/$ $-8 \times 10^{-8}$ | 68.7/79.0 | 655/2229 | 0.02/0.19 | 0.05/0.02 | <0.001/<0.001 |
| Equation (3): Height = f(age) | <0.001 | $10^{-13}/-10^{-11}$ | 78.3/66.2 | 223/766 | 0.16/0.14 | 0.01/0.15 | 0.948/0.243 |
| Equation (4): DBH = f(height) | <0.001 | 0 | 85.7/79.1 | 1500 | 0.73/0.70 | 0.63/0.60 | 0.008/0.009 |
| Equation (5): Height = f(DBH) | <0.001 | 0 | 85.7/80.0 | 772 | 0.01/0.01 | 0.18/0.19 | 0.44/0.53 |

## 4. Discussion

The nested repeat measurements linear mixed model revealed the importance of the management objectives, as more than one shoot on a stem exhibited larger slenderness coefficients, meaning they are suitable for handmade objects, whereas single-stem growing provided a faster overall growth. Our findings were confined to trees aged less than 240 months, where no asymptote was reached for the seed-managed trees and an incipient plateau was hinted at for the trees with multiple shoots in terms of height but not necessarily in terms of DBH. In fact, for the upper quartile, the single- and double-shoot trees did not reach the senescence stage, which was characterized by their reduced growth. Therefore,

our results, although powerful in terms of their findings, should be considered to be restricted by the age of the trees.

The DBH and height models for the single-shoot and seed-managed trees clearly showed a divergence in growth, irrespective the attribute, i.e., DBH or height. For the same DBH, unequivocally presented the slower height growth of the seed-managed trees compared with the single-shoot trees. Interestingly enough, the DBH growth was larger for the seed-managed trees than for single-shoot trees with respect to the height. However, the single-shoot trees had a height growth that was more than 30% larger than the seed-managed trees with respect to age (i.e., a slope of 3.0 vs. 2.3, respectively). A more nuanced picture was painted for the DBH, which had a smaller increase for the single-shoot trees (i.e., a slope of 15.1 vs. 27.2) and penalty in terms of age (i.e., a slope of $-6.9$ vs. $5.2$) but a smaller intercept (i.e., $-0.7$ vs. $-9.7$) than the seed-managed trees. Therefore, the seed-managed trees had a slower start, as can be seen in Figure 4, but once the growth adjusted to the surroundings, they overtook the shoots in term of the DBH, with this event occurring at around 15 years (i.e., 180 months) of age.

Undoubtedly, there are more accurate yield models than the two that we proposed for the DBH and height. Nevertheless, the gain inaccuracy and precision, and likely in robustness, does not justify the selection of more complex models for at least two reasons: first, the short range in terms of age, particularly with the absence of an asymptote, points towards a temporary model, and a more complete one should be developed if the measurements were to include older ages. The second reason is due to the required parsimony of any model, which as they are presented, is the simplest (i.e., three parameters).

A popular model in use in the region studied is based on the double logarithmic equation of Giurgiu [22], which has been proven to provide operational values for many species, including sycamore maple. There are three issues with the existing model that the present models do not exhibit. First, contrary to Equations (4) and (5), the present model does not provide any assessment metrics, such as coefficients of determination, AIC, or even a standard deviation; just the model is provided. Furthermore, the authors used the same function for more than 30 species, which suggests that their relationship is unsuitable for many species. Additionally, double logarithmic models that use DBH and height are prone to collinearity issues, such as the fact that log(DBH) and log(height) are highly correlated, and the model includes not only their sum but also the sum of their squares. Our model on the other hand is so parsimonious that collinearity tests are not needed (i.e., the model contains only one predictor variable). The last issue of the Giurgiu model is the reliance on the measurements being executed on a short interval, not repeatedly, as we did. Therefore, the results are not necessarily accurate, as the time variable was replaced with space to accommodate multiple ages. Consequently, the existing models should only be considered in applications for ages larger than 20, as the present formulations were correct (repeated measurements), parsimonious and were developed using standard assessment metrics.

The temporary character of the yield models for the DBH and height was confirmed by the relationship between the age and the DBH, which exhibited a clear linear relationship, irrespective the management style (coppice with one shoot or seed-managed). It should be noted that for younger ages the seed-managed trees exhibited more variability than the coppice ones, as the coefficient of correlation was almost 5% more for coppice, irrespective of the attribute (DBH or height). Therefore, for ages less than 21 years (i.e., 252 months) for seed-managed trees and 18 years (i.e., 216 months) for single-shoot coppices, the relationships for yield, DBH and height, could be used with confidence, but extrapolation outside the predictive range is highly not recommended.

## 5. Conclusions

Sycamore maples are trees with significant economical, ecological and cultural value. Their presence is widespread in wood manufacturing, they contribute to ecosystem diversity and site productivity and they are also encountered in many folklore tales. Even though it covers most of the sustainable forest management aspects in a profound manner,

its investigation does not mirror its importance. Therefore, we focused on answering three research questions. The first question aimed at finding the most suitable coppice strategy for achieving ground coverage and biomass. The second and third questions concentrated on providing numerical solutions to the development of sycamore maple trees, namely the yield and the dimensional relationships (i.e., DHB vs. height and vice versa). Using a series of eight measurements spanning twenty-one years (i.e., two hundred and fifty-two months), starting from age six (i.e., seventy-two months), we found evidence that the single-shoot coppices provided superior yields than the seed-managed trees up to age twelve (one hundred and forty-four months) in terms of height and up to age twenty (two hundred and forty months) for DBH. However, the coppice trees clearly outperformed the seed trees up to age 10 (the end of the measurements for height due to the dense canopy). Given the reduced time range and the concentration on one location (i.e., the similar productivity and virtually identical climate), the yield of the DBH and height for the single-shoot trees and the seed-managed trees were described by simple formulations, namely the Schumacher model for the DBH and square root for the height. The parsimonious yield models suggested the usage of simple formulas to represent the development of young plantations. In addition, the relationship DBH–height exhibited a clear linear form, pointing towards the main limitation of the study, namely the confinement to ages less than 20 years (i.e., 240 months). The presence of the models with no asymptote for the DBH–height models confirmed the invertibility of the two relationships but pointed towards the reduced range of ages. Nevertheless, the richness of the information and the accuracy of the models developed in the present study recommend their usage in research and practical applications. We expect to continue the measurements for at least 10 more years, at which point we assume that the DBH asymptote will have been reached, triggering the development of a comprehensive model for sycamore maple trees.

**Author Contributions:** Conceptualization, V.-N.N.; methodology, B.M.S.; software, B.M.S.; validation, V.-N.N.; formal analysis, B.M.S.; resources, V.-N.N. and B.M.S.; writing—original draft preparation, B.M.S. and V.-N.N.; writing—review and editing, B.M.S. and V.-N.N.; visualization, V.-N.N. All authors have read and agreed to the published version of the manuscript.

**Funding:** The study was partially funded by the USDA grant McIntire Stennis project OREZ-FERM-907.

**Data Availability Statement:** Not applicable.

**Acknowledgments:** We wish to thank Wilhelm Sandi and Melinda Sandi-Szaboné, as well as some of our B.Sc. class students, for their help during the establishment and measurement stages of our work.

**Conflicts of Interest:** The authors declare no conflict of interest.

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
