# Peer review of "Coppice Management for Young Sycamore Maple (Acer pseudoplatanus L.)"

_forests, doi:10.3390/f14020297_

Round 1

Reviewer 1 Report

There is no doubt about the suitability of the paper for the journal. The weak point of the paper is the layout of the text and text editing, which has serious drawbacks. That attracts the attention of the reader much more than the content of the article itself. It must be adjusted. I do not think references are cited according to the MDPI style (in the list of references).

Suggestion: I would not repeat the key words from the title – maybe to consider the title: Coppice management for young Acer pseudo-2 platanus L. and „sycamore maple“ remove to the key words.

Author Response

We would like to thank to the two reviewers for their time and efforts on improving our manuscript. We would also like to thank them for the positive decision and the constructive feedback. The present Rebuttal Letter details all the changes that we executed to the manuscript to address their comments. The Letter is structured such that each comment is individually addressed. For an easy read of the Letter we identified the reviewer comments by italics and our answers by normal font.        

Reviewer#1

The reviewer has a positive view on or paper, which we thank. The reviewer also has a reduced number of suggestions, namely one (below).

I would not repeat the key words from the title – maybe to consider the title: Coppice management for young Acer pseudoplatanus L. and „sycamore maple“ remove to the key words.

We have adjusted the keywords to reflect the reviewer suggestion.

the conclusion, which now is different than the abstract.

Reviewer 2 Report

General overview

The study describes the growth of shoots from stumps (measurement of DBH and height) compared to individuals of generative origin of sycamore maple. The need to describe the process of tree growth of this tree species in coppice forest is certainly high. But the manuscript has a large number of shortcomings: Unclear wording and expressions, unclearly arranged text and images, insufficiently explained methodological approaches. Discussion and Conclusion have very insufficient content.

The experiment seems unprepared, especially the data collection. It looks like the autors are measuring shoots from stumps at thoughtless random times. The study includes data from only one location and the set of sampled stem is quite low - shoots from 40 stumps . The chosen word expressions in the text cause ambiguity. It is necessary to replace them with scientific terms. There are many long sentences in the text, it is necessary to divide them into simple sentences. Because in most parts the text and message is unclear and there are a lot of inaccuracies, I was not able to point out all the flaws.

The manuscript will need to be revised from the ground up, the choice of methods must be thoroughly justified and some chapters will need to be finished.

Introduction

L 52 – “intimate mixtures” – In my opinion, this term is quite far from forestry terminology. What do you imagine under that?

L 61-63 “It coppices“ – Coppice means undergrowth. In my opinion, it is not the same as regenerating, creating shoots. Alternatively, please explain. “…growth…leading“ – why isn't there “growth… leads“?

L 64-65 “used in coppice stands“ is a different meaning than „maple trees… coppice well”. In the first case it is a type of forest stand, in the second case a verb?

L 69-70 “The species grows fast in height up to 20-25 years“ – This sentence repeats what was said in the previous sentence. Please delete it, or edit the previous sentence.

L 71 “Claessens et al. 1999, in [17])“ – this method of citation is very unusual and, in my opinion, unnecessary. It is always necessary to go to the original source (even though you may not have it directly in your hands) and cite it. So add this source Claessens et al. 1999 to References

L 71-72 “it falls off considerably“ – This sentence is so complicated that I'm not sure what exactly "it" falls off? Height growth? Better replace with a specific term. And the term "decrease" would be better.

L 76 Same case as on L 71

Material and Methods

It is necessary to establish a paragraph in which the terms would be explained. E.g. seed-generated, high forest, shoot, stem, …

L 91-92 The sentence is too complicated, the part "numerical support in the selection of a particular coppice strategy for the sycamore maple" is repeated from the end of the Introduction, please delete this part and simplify the sentence.

L 93 “sub-compartment 81E, former VI Bodoc Forest Management Unit“ especially this information is useless in my opinion, most readers probably don't know the local forest stands. Please delete.

L 94 “45o54’35’’ N. lat, 25o54’16’’ E long.“ - Lat. and long. no need to state. When coordinates are entered without lat. and long. into the web browser, the location can be found. Therefore, please delete these abbreviations. N. enter without a dot. The character “ ° ” looks more like “o” here, please verify and replace. “from Romania” – usually written “in Romania”.

L 95 “D.f.k” type. But in this classification there are only categories Dfa, Dfb, Dfc, Dfd; elsewhere "k" is referred to as "cold arid". Is it really the climate of Romania? According to the map, for example, here http://koeppen-geiger.vu-wien.ac.at/present.htm is the majority of Romania in the Dfb category.

"(i.e." - the parenthesis was not closed

L 97 “The site is relatively horizontal” – What does it mean? Like Terrain is relatively flat? “brown argillic soil” – indicate the classification used. This soil type name is not exactly common.

L 98 “for the natural vegetation (sessile oak-dominated mixture of broadleaves)“ – strange wording. Why don't you write that the natural vegetation consists of a sessile oak-dominated mixture of broadleaves?

L 99 Figure 1. This map shows the location very poorly, moreover the names of the settlements are in small unreadable font. Replace it with one with a smaller scale, where at least a larger part of Romania will be shown.

L 103-115 The content of this paragraph does not thematically belong to the subchapter 2.1 Study area

L 106 “sycamore” – everywhere else in the text you use the term "sycamore maple", so add the name here as well.

L 130, 138 – What is the significance of setting these DBH and height measurement intervals? No rule can be seen in the differences of months between the intervals, they are of different lengths. Either state what the significance is or prove that different interval lengths do not affect the structure of the data obtained? Although it is understandable that seed-generated trees started to be measured later, their interval times are often shifted compared to coppiced trees by several months? Why?

Why do you state the measurement interval in months and not in years? It is clearer to write, for example, 11.6 years. Under the number of months, it is hard to imagine how long it is.

L 135 “100sycamore maples“ – Correct on “100 sycamore maples”

L 159 “and height the number of shoots that…“ I don't understand the sentence, should it be "height of number of shoots that..."?

L 162 “We selected” – a space is missing before the sentence.

Results

L 172, 173 "As expected" - it was assumed on the basis of what? In my opinion, in Results should not be an expression of assumption.

L 184 Table 1 is confusing at first glance. “Shoots” needs to be changed to “Number of Shoots”, or “No. of Shoots”. The columns are unnecessarily wide, instead of "Maximum" it would be enough to state "Max.", instead of "Minimum" state "Min." When you list “DBH [cm]”, why is there no “[months]” next to “Age”?

L 186 Figure 4. Here on the x-axis the age is in years, so why are the months listed above in the text? "75% percentile" is a doubling of %, correct to "75th percentile". Why is the 75th percentile shown in both the top and bottom charts and the 25th percentile only in the bottom? Why is there no Mean value in the graphs?

L 190 Why is this paragraph in italics? In addition, the stated values are difficult to verify when the time intervals are given in months and the measurement intervals of shoots and seed-managed trees are different.

L 195 “DBH and height seem to be linearly related“ – What makes you think so? In Figure 5, there is no regression model showing the dependence of x and y. In my opinion, the Results cannot contain considerations of the "seems that" type, but only a description of the actual state.

L 196 “the larger the number of shoots on the stump the slower the dimensional growth” – This rating is based on Figure 4, so move it to the comments of Figure 4.

L 200 Figure 5. This graph show individuals under the age of 12? It is not mentioned in the label.

L 208, 227, 230 “stem” – unclear terminology, stem or shoot? It is necessary to unify throughout the text. It is best to create a paragraph describing the terminology.

L 231 "as they provide comparable volume" - you mention the volume of stems, but volume is not mentioned either in Material and Methods or elsewhere in Results.

L 232 Schumacher model [30] - should be listed in Material and Methods with reference.

L 249-253 This paragraph does not belong in Results, but in Introduction and Material and Methods

L 265 Table 3. It is not clear from the model names in the Model column which specific model is mentioned above. The models have inconsistent names. It needs to be unified. In the last line "019" should probably be "0.19".

Discussion

Discussion does not discuss the results with other studies. That's a big shortcoming. A comparison with any other models used in other studies is missing.

L 267-272 Again, why is the text in italics?

L 282-284 Numbers in parentheses do not have units. It is not clear what they are expressing. "penalty of the age" - it is not clear what is meant by this.

L 285 "seed-managed trees have a slower start" - how can you evaluate the start of growth of seed-managed trees, when you only started measuring them from 9 years of age?

Conclusion

The text is exactly the same as in the Abstract. This is unacceptable.

Author Response

We would like to thank to the two reviewers for their time and efforts on improving our manuscript. We would also like to thank them for the positive decision and the constructive feedback. The present Rebuttal Letter details all the changes that we executed to the manuscript to address their comments. The Letter is structured such that each comment is individually addressed. For an easy read of the Letter we identified the reviewer comments by italics and our answers by normal font.        

 Reviewer#2

The reviewer has a positive view on our paper, which we also thank. The reviewer spent a significant effort trying to improve our manuscript, which we appreciate, as provided a many suggestions (below).

The study describes the growth of shoots from stumps (measurement of DBH and height) compared to individuals of generative origin of sycamore maple. The need to describe the process of tree growth of this tree species in coppice forest is certainly high. But the manuscript has a large number of shortcomings: Unclear wording and expressions, unclearly arranged text and images, insufficiently explained methodological approaches. Discussion and Conclusion have very insufficient content.

We have executed major changes to the entire manuscript, which is now more fluent, contains more details, and has the figures and text better aligned.

The experiment seems unprepared, especially the data collection. It looks like the authors are measuring shoots from stumps at thoughtless random times. The study includes data from only one location and the set of sampled stems is quite low - shoots from 40 stumps. The chosen word expressions in the text cause ambiguity. It is necessary to replace them with scientific terms. There are many long sentences in the text, it is necessary to divide them into simple sentences. Because in most parts the text and message is unclear and there are a lot of inaccuracies, I was not able to point out all the flaws.

The manuscript was re-written such that explains better the reason for sampling at irregular intervals, mainly because of economic constraints, particularly measurements funding restrictions by the landowner. We also replaced and correct the loose vocabulary and language inaccuracies.

The manuscript will need to be revised from the ground up, the choice of methods must be thoroughly justified, and some chapters will need to be finished.

We have thoroughly revised the manuscript.

L 52 – “intimate mixtures” – In my opinion, this term is quite far from forestry terminology. What do you imagine under that?

We changed the structure of the paragraph, and now the part that was referred to “intimate mixture” is “integrated with other tree species”.

L 61-63 “It coppices“ – Coppice means undergrowth. In my opinion, it is not the same as regenerating, creating shoots. Alternatively, please explain. “…growth…leading“ – why isn't there “growth… leads“?

We have transformed significantly the introduction, and now the part with “it coppices” does not exist anymore. We also changed the “leading” to “leads” as it is more appropriate in the new context, which now is like:

“The rapid height growth combined with the large number of shoots leads to canopy closure occurring in the second or third year after coppicing (Evans 1984).”

L 64-65 “used in coppice stands“ is a different meaning than „maple trees… coppice well”. In the first case it is a type of forest stand, in the second case a verb?

As a part of the rewrite, we have eliminated the term of “coppice stands” as a prt fo a new sentence which now reads as: “The high yield of the young shoots promoted the copse, as more than 2500 ha are present in Britain which experience a rotation of 10-20 years – Evans 1984).”

L 69-70 “The species grows fast in height up to 20-25 years“ – This sentence repeats what was said in the previous sentence. Please delete it, or edit the previous sentence.

We have eliminated the duplicated portion of the sentence and adjust the remaining to reflex the flow of the narrative: “Beyond age 50 years the height growthdecreases considerably (Joyce et al. 1998; Hein et al. 2009”

L 71 “Claessens et al. 1999, in [17])“ – this method of citation is very unusual and, in my opinion, unnecessary. It is always necessary to go to the original source (even though you may not have it directly in your hands) and cite it. So add this source Claessens et al. 1999 to References

Thank you for the recommendation. We added the reference.

L 71-72 “it falls off considerably“ – This sentence is so complicated that I'm not sure what exactly "it" falls off? Height growth? Better replace with a specific term. And the term "decrease" would be better.

 We replace the syntagm with “decrease”, as recommended.

L 76 Same case as on L 71

We rechecked the sentence and in its current for fits the narrative, as it discusses similar statistics for DBH as we did for height.

Material and Methods

It is necessary to establish a paragraph in which the terms would be explained. E.g. seed-generated, high forest, shoot, stem, …

We changed the introduction, such that all terms are explained before the methods.

L 91-92 The sentence is too complicated, the part "numerical support in the selection of a particular coppice strategy for the sycamore maple" is repeated from the end of the Introduction, please delete this part and simplify the sentence.

We have changed the sentence which now looks like “To support the finding of our study we have carried out an experiment located…”

L 93 “sub-compartment 81E, former VI Bodoc Forest Management Unit“ especially this information is useless in my opinion, most readers probably don't know the local forest stands. Please delete.

We agree with the reviewer, that those details are irrelevant. We delted them

L 94 “45o54’35’’ N. lat, 25o54’16’’ E long.“ - Lat. and long. no need to state. When coordinates are entered without lat. and long. into the web browser, the location can be found. Therefore, please delete these abbreviations. N. enter without a dot. The character “ ° ” looks more like “o” here, please verify and replace. “from Romania” – usually written “in Romania”.

We implemented the reviewer’s comments, as they will ensure a consistent narrative.

L 95 “D.f.k” type. But in this classification there are only categories Dfa, Dfb, Dfc, Dfd; elsewhere "k" is referred to as "cold arid". Is it really the climate of Romania? According to the map, for example, here http://koeppen-geiger.vu-wien.ac.at/present.htm is the majority of Romania in the Dfb category.

The Dfk is the correct climate, as it represents all the climate parameters from the Koppen system using the local conditions, not the regional values.

"(i.e." - the parenthesis was not closed

We closed the parenthesis, as indeed the series was not complete.

L 97 “The site is relatively horizontal” – What does it mean? Like Terrain is relatively flat? “brown argillic soil” – indicate the classification used. This soil type name is not exactly common.

We have changed the sentence to me more specific and added the proper references for the soil classification system. The new sentence is “The site is relatively horizontal, with slope less than 5°, and has a brown argillic soil[6,7], …”

L 98 “for the natural vegetation (sessile oak-dominated mixture of broadleaves)“ – strange wording. Why don't you write that the natural vegetation consists of a sessile oak-dominated mixture of broadleaves?

We changed the sentence as recommended by the reviewer. The new sentence is now: “withmoderate fertility for the natural vegetation consisting of a sessile oak-dominated mixture of broadleaves”.

L 99 Figure 1. This map shows the location very poorly, moreover the names of the settlements are in small unreadable font. Replace it with one with a smaller scale, where at least a larger part of Romania will be shown.

We redid the figure as recommended by the reviewer.

L 103-115 The content of this paragraph does not thematically belong to the subchapter 2.1 Study area

We have changed the title of the subchapter to “Study Area and experiment establishment” , to capture the content of the narrative within the subchapter.

L 106 “sycamore” – everywhere else in the text you use the term "sycamore maple", so add the name here as well.

We changed “sycamore” with “sycamore maple”, as suggested.

L 130, 138 – What is the significance of setting these DBH and height measurement intervals? No rule can be seen in the differences of months between the intervals, they are of different lengths. Either state what the significance is or prove that different interval lengths do not affect the structure of the data obtained? Although it is understandable that seed-generated trees started to be measured later, their interval times are often shifted compared to coppiced trees by several months? Why?

Weagreewith the reviewer that uneven measurements are not preferred because they introduce an extra element of randomness in the analysis. We have added two paragraphs, one for copse and one for seed-trees, explain the reason for uneven measurement intervals.

“The irregular acquisition of data, almost annually for the first six times, and more than two years for the last two measurements has two main reasons. First, the landowner focus was not on the trees but on the annual crop; therefore, all the measurements were executed to accommodate the main land management objectives.  While the trees were relatively short, they were measured every year when the weather permitted. The last two measurements were carried out opportunistically because there were no annual crops and external funding to support or justify the field effort.”

Why do you state the measurement interval in months and not in years? It is clearer to write, for example, 11.6 years. Under the number of months, it is hard to imagine how long it is.

We agree with the reviewer that expressing the age in moth is harder to conceptualize, but given the reduced number of measurements and the lack of replicated sites we decided to eliminate other sources of variation, such as rounding errors.

L 135 “100sycamore maples“ – Correct on “100 sycamore maples”

We corrected as suggested by the reviewer.

L 159 “and height the number of shoots that…“ I don't understand the sentence, should it be "height of number of shoots that..."?

We thank the reviewer for catching the missing parts of the sentence. The new formulation is “… height for shoots that provided the fastest developments, namely the stumps with one shoot”

L 162 “We selected” – a space is missing before the sentence.

We have added the space, as suggested.

L 172, 173 "As expected" - it was assumed on the basis of what? In my opinion, in Results should not be an expression of assumption.

We change the sentence to reflect the point of the reviewer. The new sentence is now “Similarly to other studies [Nicolescu et al 2013, Nicolescu et al 2018], the DBH disparity … ”

L 184 Table 1 is confusing at first glance. “Shoots” needs to be changed to “Number of Shoots”, or “No. of Shoots”. The columns are unnecessarily wide, instead of "Maximum" it would be enough to state "Max.", instead of "Minimum" state "Min." When you list “DBH [cm]”, why is there no “[months]” next to “Age”?

We made the changes suggested by the reviewer, except abbreviate the statistics, which are not necessarily clear for all readers.

L 186 Figure 4. Here on the x-axis the age is in years, so why are the months listed above in the text? "75% percentile" is a doubling of %, correct to "75th percentile". Why is the 75th percentile shown in both the top and bottom charts and the 25th percentile only in the bottom? Why is there no Mean value in the graphs?

We have changed the figure as recommended. The mean, first and third percentiles were not all on the same plot to keep the pictures clean and easy to understand. Adding eight extra lines would have clutter the figures beind understanding.

L 190 Why is this paragraph in italics? In addition, the stated values are difficult to verify when the time intervals are given in months and the measurement intervals of shoots and seed-managed trees are different.

It was an error that appeared during the conversion from Word to Adobe pdf. We correct it now.

L 195 “DBH and height seem to be linearly related“ – What makes you think so? In Figure 5, there is no regression model showing the dependence of x and y. In my opinion, the Results cannot contain considerations of the "seems that" type, but only a description of the actual state.

The plot of data suggests the type of relationship suitable for modeling two or more variables. Among all the relationship, the linear is the most robust, in the sense that it ensures the preservations of many properties across rings or ideals, particularly isomorphism. The formalization of the relationship comes after the type is selected and fit. At this stage, we just want to point the fact that a linear relationship is suitable for representing DBH-height relationship. Later, we provide evidence that our presumption is correct.

L 196 “the larger the number of shoots on the stump the slower the dimensional growth” – This rating is based on Figure 4, so move it to the comments of Figure 4.

The inference is based on Fig. 5, as it describes the DBH-height relationship, whereas Fig 4 present the change in DBH and height thru time.

L 200 Figure 5. This graph show individuals under the age of 12? It is not mentioned in the label.

We have added the less than 12 years into the caption of the figure, as suggested by the reviewer.

L 208, 227, 230 “stem” – unclear terminology, stem or shoot? It is necessary to unify throughout the text. It is best to create a paragraph describing the terminology.

We have added a sentence in the methods that explicitly states what shoots or stems means. The sentence is “ Depending on the context, stems could refers to shoots or seed-trees”.

L 231 "as they provide comparable volume" - you mention the volume of stems, but volume is not mentioned either in Material and Methods or elsewhere in Results.

We replace “volume” with “dimensions” to maintain the consistency of the discourse.

L 232 Schumacher model [30] - should be listed in Material and Methods with reference.

We have added the reference

L 249-253 This paragraph does not belong in Results, but in Introduction and Material and Methods

We have significantly altered the paragraph, as recommended by the reviwer, which now look like “We developed two inverse relationships for DBH and height, because various applications requires either prediction of height from DBH or the reciprocal. Nevertheless, the functions are not mathematically the inverse of each other, because both relationships have an asymptote with each other. The need of two relationships is also advocated by the dependency of ….”

L 265 Table 3. It is not clear from the model names in the Model column which specific model is mentioned above. The models have inconsistent names. It needs to be unified. In the last line "019" should probably be "0.19".

We have added more details to the table which , now is better integrated with the narrative.We have changed the 019 to 0.19, as reviewer correctly pointed.

Discussion does not discuss the results with other studies. That's a big shortcoming. A comparison with any other models used in other studies is missing.

Because there are only few studies focused only on sycamore maple we have a short discussion. Nevertheless, we have added a paragraph comparing our results with other studies that exists in the region. The addition is

“A popular model in use in the region is based on the double logarithmic equation of Giurgiu [2], which is proven to provide operational values for many species, including sycamore maple. There are three issues with the existing model that the present models does not exhibit. First contrary to the Eq. 4 and 5, the present model does not provides any assessment metrics, such as coefficient of determination, AIC, or even a standard deviation, just the model. Furthermore, the authors used the same function for more than 30 species, which suggests that the relationship is unsuitable for many species. Additionally, the double logarithmic models that uses DBH and height is prone to colinearly issues, as log(DBH) and log(height) are highly correlated, and the model includes not only their sum bust also the sum of their squares. Our model on the other hand, is so parsimonious that collinearity tests are not needed (i.e., contains only one predictor variable). The last issue of the Giurgiu model, is the reliance of measurements executed on a short interval, not repeatedly, as we did. Therefore, the results are not necessarily accurate, as the time was replaced with space to accommodate multiple ages. Consequently, the existing models should be considered in applications only for ages larger than 20, as the present formulations are correct (repeated measurements), parsimonious, and are developed using the standard assessment metrics.”

L 267-272 Again, why is the text in italics?

The italics should not be there, most likely is appearing during the conversion for Word to pdf. I assume that it would be fixed during the editorial process.

L 282-284 Numbers in parentheses do not have units. It is not clear what they are expressing. "penalty of the age" - it is not clear what is meant by this.

The values in parenthesis have units, depending on the relationship, but because they are coefficients in a regression usually the units are not display. Nevertheless, we have specified in text the fact that the values are parameters not some summary statistics).

L 285 "seed-managed trees have a slower start" - how can you evaluate the start of growth of seed-managed trees, when you only started measuring them from 9 years of age?

It can be inferred from the parameters of the equations 2 and 3 and figure 4, as the dimensions of seed -trees are smaller than the dimensions of the single shoots for ages less than 15 years. We have changed the sentence to make it more fluent and clearer: “Therefore, the seed-managed trees have a slower start, as can be seen in Figure 4, but once the growth adjusts to the surroundings,…”.

The text is exactly the same as in the Abstract. This is unacceptable.

We enlarged the conclusion, which now is different than the abstract.

Round 2

Reviewer 2 Report

Dear authors,

I think the manuscript has been quite extensively edited and improved by the authors, and most of my comments from version 1 have either been accepted or, if not, adequately explained. I have included my comments in the version of the manuscript resulting from the comparison of the original and the revision 1. This manuscript is in change mode, it shows the changes made by the authors between version 1 and 2. Hope this is not a problem for you.

There are only a few of my comments and they mostly point to imperfections already appearing in version 1 and uncorrected by the authors.

Author Response

Rebuttal Letter

We appreciate the thoroughness, patience, and diligence of the reviewer. The manuscript is in the present form is much superior compared with the initial submission. Similarly to the first rebuttal letter, we have included the reviewer comments in italics, which are followed by our response in normal font.

Inconsistency in quotes.

We changed the quotation marks from ‘ to “, as suggested. Thank you for pointing this out.

In the Cover letter you write: "We have added two paragraphs, one for coppice and one for seed-trees, explaining the reason for uneven measurement intervals." But where are the paragraphs?

The reviewer is correct. We forgot to put the paragraphs inside the manuscript; we left it only in the rebuttal letter. Thank you for pointing this error. WE have added the paragraphs inside the manuscript now. They are:

“The irregular acquisition of data, almost annually for the first six times, and more than two years for the last two measurements has two main reasons. First, the landowner focus was not on the trees but on the annual crop; therefore, all the measurements were executed to accommodate the main land management objectives.  While the trees were relatively short, they were measured every year when the weather permitted. The last two measurements were carried out opportunistically because there were no annual crops and external funding to support or justify the field effort.”

“The same argument for unequal measurements presented for the shoots is valid for the seed-managed trees.” But there is no such argument in the text.

We have added the paragraph to the shoots that clarifies the reasons for uneven measurements.

You argue that converting units to years would result in a rounding error. That is possible. But why is Table 1 in units of months, while Figure 4 is then in units of years? This is an incomprehensible inconsistency of units of time. In the text after Table 1, you describe the time variable only in years, not in months. Please unify time units throughout the manuscript.

We changed the “years” to “months” in Figure 4 and its caption, as suggested

Comparing the data between Table 1, Figure 4 and the text in the Results requires the time values to be constantly divided and multiplied by 12 (the number of months in the year). Unify time units throughout the manuscript.

We have changed the units from “years” to months” throughout the manuscript when warranted.

In the Conclusion, a large proportion of sentences remain identical to the Abstract. The Conclusion should emphasize the interesting findings of the study and not just repeat the Abstract. But the quality of your published article is up to you.

We have added more details to Conclusions, which now includes the following sentences:

“The parsimonious yield models suggest the usage of simple formulas to represent the development of young plantations.  […]. The presence of models with no asymptote for the DBH-height models confirms the invertibility of the two relationships but points towards the reduced range of ages.”